# Genome-wide selection inference at short tandem repeats

**Bonnie Huang**[1], **Arun Durvasula**[2,3], **Nima Mousavi**[4], **Helyaneh Ziaei-Jam**[5], **Mikhail Maksimov**[6], **Kirk E. Lohmueller**[2,3☉*], **Melissa Gymrek**[5,6☉*]

**1** Department of Bioengineering, University of California San Diego, La Jolla, California, United States of America, **2** Department of Ecology and Evolutionary Biology, University of California Los Angeles, Los Angeles, California, United States of America, **3** Department of Human Genetics, David Geffen School of Medicine, University of California Los Angeles, Los Angeles, California, United States of America, **4** Department of Electrical and Computer Engineering, University of California San Diego, La Jolla, California, United States of America, **5** Department of Computer Science and Engineering, University of California San Diego, La Jolla, California, United States of America, **6** Department of Medicine, University of California San Diego, La Jolla, California, United States of America

☉ These authors contributed equally to this work.
* klohmueller@g.ucla.edu (KEL); mgymrek@ucsd.edu (MG)

## Abstract

Short tandem repeats (STRs) comprising repeated sequences of 1–6 bp are one of the largest sources of genetic variation in humans. STRs are known to contribute to a variety of disorders, including Mendelian diseases, complex traits, and cancer. Based on their functional importance, mutations at some STRs are likely to introduce negative effects on reproductive fitness over evolutionary time. We previously developed SISTR (Selection Inference at STRs), a population genetics framework to measure negative selection against individual STR alleles. Here, we extend SISTR to enable joint estimation of the distribution of selection coefficients across a set of STRs. This method (SISTR2) allows for more accurate analysis of a broader range of STRs, including loci with low mutation rates. We apply SISTR2 to explore the range of feasible mutation parameters and demonstrate substantial variation in mutation and selection parameters across different classes of STRs. Finally, we estimate the relative burden of *de novo* and inherited variation at STR vs. single nucleotide variants (SNVs). Our results suggest that whereas SNVs contribute a greater total burden of inherited variation in a typical genome, the burden of *de novo* mutations at STRs is greater than that of SNVs. Overall, we anticipate that the evolutionary insights gained from this study will be important for future studies of variation at STRs and their role in evolution and disease.

**Data availability statement:** Data availability Analysis is based on data from the 1000 Genomes Project available at SRA accession PRJEB31736. Summary statistics used for inferences here are available on Figshare https://doi.org/10.6084/m9.figshare.30261118. Code availability SISTR2 source code and documentation can be found at: https://github.com/BonnieCSE/STRSelection and https://github.com/BonnieCSE/SISTR2.

**Funding:** This study was supported by the Simons Foundation Autism Research Initiative (https://www.sfari.org/) grant #630705 (M.G.), the National Human Genome Research Institute, National Institutes of Health (https://www.genome.gov/) grant R01HG010149 (M.G.) and National Institutes of Health Common Fund (https://commonfund.nih.gov/) award U01HG013442 (M.G.). M.G. was additionally supported in part by the Office Of The Director, National Institutes of Health (https://www.nih.gov/institutes-nih/nih-office-director) under Award Number DP5OD024577. K.E.L. was supported by the National Institutes of Health (https://www.nih.gov/) grant R35GM119856. The funders had no role in study design, data collection and analysis, decision to publish, or preparation of the manuscript.

**Competing interests:** The authors have declared that no competing interests exist.

## Author summary

Short tandem repeats (STRs), which consist of repeated sequences of 1–6 bp, often show variation in repeat copy number across individuals. Despite STRs being a significant source of genetic variation in humans, the evolutionary forces acting on them are largely unknown. Here we develop a new computational approach, called SISTR2, to estimate the deleterious effects of STR mutations. Application of SISTR2 to STRs throughout the human genome reveals that many STR variants are predicted to be deleterious, with the specific effects depending on the repeat length, sequence, and location in the genome. When comparing the fitness effects of STR variants to those of single nucleotide variants (SNVs), we find that inherited SNVs have a larger total effect on fitness than STRs. On the other hand, when considering new mutations in a single generation, the pattern is reversed, with the average burden of new deleterious mutations at STRs being greater than that of SNVs, driven primarily by higher mutation rates at STRs. Our work suggests that STRs are a significant source of deleterious variation in the human genome.

## Introduction

Short tandem repeats (STRs) are DNA sequences consisting of repeated 1–6 base pair motifs that comprise approximately 1.6 million loci in the human genome [1]. Due to their high prevalence in the genome and rapid mutation rates, variation in copy number at STRs represents a large portion of human genetic variation. Recent evidence supports a role for STRs in diverse biological processes that control gene regulation [2,3] and contribute to a wide range of human traits [4]. Based on their functional importance, mutations at some STRs are likely to introduce detrimental effects on reproductive fitness. Understanding these fitness effects can provide insights into the role of STRs in evolution.

Previous studies have used multiple approaches to measure the effects of natural selection on STRs. Haasl and Payseur developed a detailed model of STR evolution including mutation, genetic drift, and natural selection at an STR implicated in Friedreich's Ataxia [5]. They recently applied a similar model to 94 STRs genotyped in eight human populations, finding evidence of natural selection on 43 STRs and a range of fitness effects [6]. However, fitting this model is computationally intensive due to the large number of parameters, making it infeasible to fit individually at each of the more than one million STRs in the genome. We previously developed an STR constraint metric based on comparing observed vs. expected mutation rates [7]. This metric could broadly distinguish neutrally evolving STRs from those implicated in severe early-onset disorders. However, that score is based on noisy STR mutation rates that are computationally expensive to estimate from individual-level genotypes, does not model the known dependence of mutation rate on allele length, and only produces locus-level, rather than allele-level scores.

Recently, we introduced SISTR (Selection Inference at Short Tandem Repeats), a computationally efficient method to measure negative selection at STRs [8]. SISTR estimates per-locus selection coefficients by finding the selection parameters that best fit the allele length distribution for one STR at a time. These fine-grained scores enable predicting the fitness impact of individual alleles at a specific locus. However, this approach faces several limitations. First, it has low power to detect weak selective effects. Furthermore, at STRs with extremely low mutation rates, such as short trinucleotide repeats, it is unable to precisely estimate selection coefficients since low levels of genetic variation could be caused by a low mutation rate, strong negative selection, or a combination of both forces.

Here, we extend SISTR to enable joint estimation of the distribution of selection coefficients across a set of STRs. This method analyzes standing variation in the population and allows us to more accurately analyze a broader range of STRs, including loci with low mutation rates. We apply SISTR2 to explore the range of feasible mutation parameters and demonstrate substantial variation in mutation and selection parameters across different classes of STRs. Finally, we show that *de novo* STR mutations tend to confer a larger total genome-wide selective burden compared to *de novo* single nucleotide mutations in either coding regions or conserved noncoding regions. However, for inherited variants segregating in a typical genome, single nucleotide variants (SNVs) convey a greater total fitness burden compared to STRs. Overall, we anticipate that the evolutionary insights gained from this study, including a more detailed understanding of STR mutation and selection parameters for different types of STRs, will be important for future studies of variation at STRs and their role in evolution and disease.

## Results

### Overview and validation of the SISTR2 joint inference method

We previously developed SISTR, a population genetics framework that estimates selection coefficients at individual STRs [8]. SISTR incorporates an evolutionary model of STR variation that includes mutation, negative natural selection, and genetic drift. Our mutation model is based on a generalized stepwise mutation (GSM) model with two modifications, including a length-dependent mutation rate and a directional bias in mutation sizes toward an optimal (central) allele length (Fig 1a; **Methods**). To model negative selection, we assume a symmetric model in which the optimal allele at each STR has a fitness of 1, and the fitness of other alleles decreases linearly with their absolute distance, in number of repeat units, from the optimum. The decrease in fitness of nonoptimal alleles scales with $s$, which ranges from 0 (no effect on fitness) to 1 (any genotype in which both alleles differ from the optimum is lethal).

Since the optimal allele is typically unknown, for most applications we assume the modal allele in the population is the optimal allele and treat these interchangeably. Indeed, recent work has suggested that the modal allele is the most influential summary statistic at inferring the optimal allele and in cases where it was possible to infer the optimal allele, it was typically the modal allele (in single optima models) or a multiple of the optimal allele (in periodic optima models) [6]. Further, we carried out simulations based on the mutation and selection models used here and found that as long as selection is at least modest, the modal allele is in most cases equal to or centered around the optimum (S1-S3 Figs). An exception is loci with relatively long optima under weak selection, for which the mode is frequently lower than the optimum (S2 Fig), a trend driven by length-dependent mutation rates [9] (S3 Fig). We evaluate the robustness of our method to errors in inferring the optimal allele below.

SISTR leverages a previously developed technique [5] that incorporates mutation, selection, and demographic models (**Methods**) to simulate allele frequencies forward in time (S3 Fig). Using approximate Bayesian computation (ABC), we determine the posterior distribution of $s$ at each locus by comparing observed allele frequencies to those simulated by our model. We previously showed that SISTR performs well on STRs with high mutation rates but is underpowered to detect selection at STRs with low mutation rates or under only modest selection. In those cases, information contained in genetic variation at a single locus is insufficient to accurately infer selection.

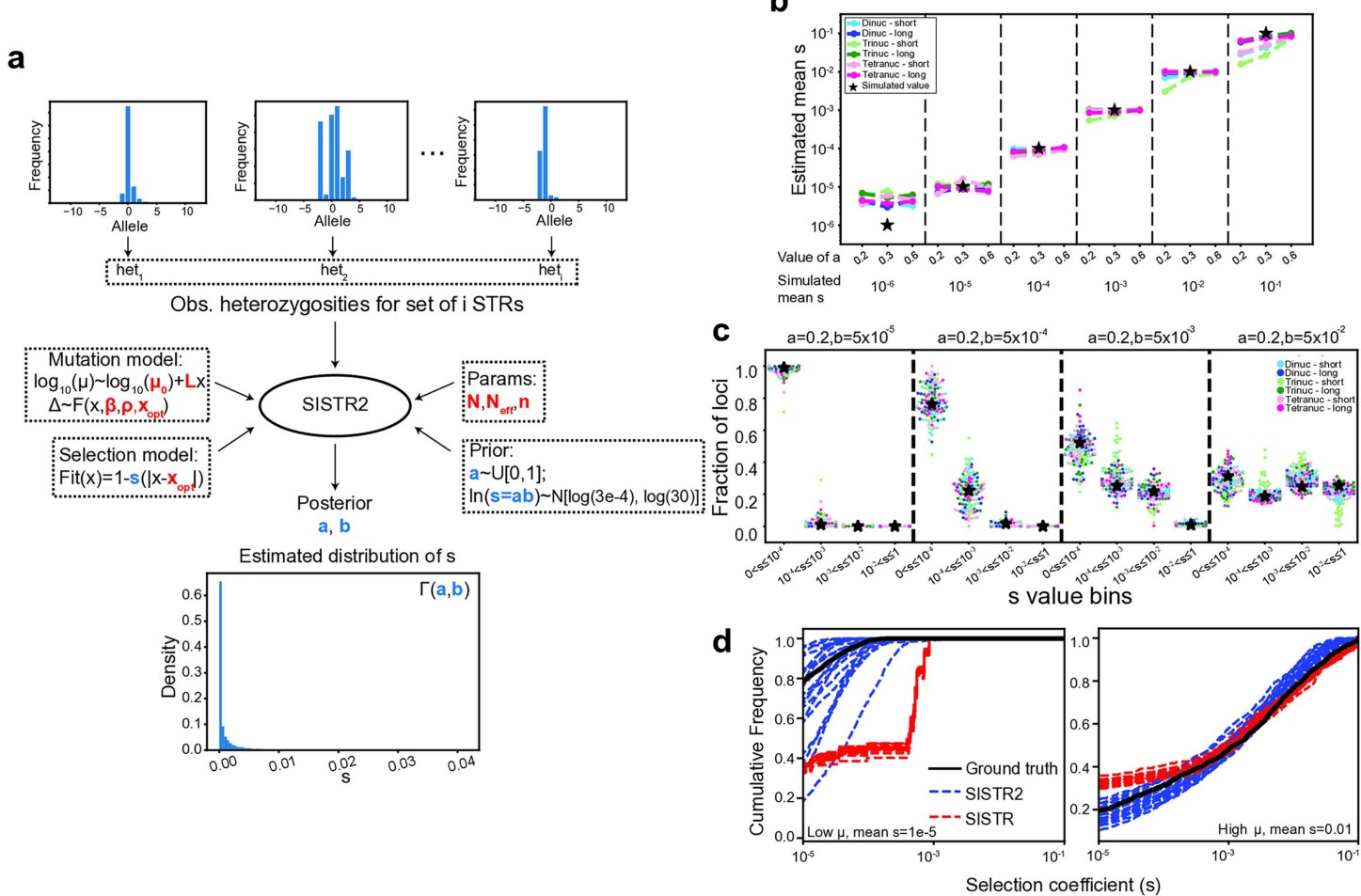

**Fig 1. SISTR2 estimates the distribution of fitness effects (DFE) across a set of STRs.** (a) Overview of SISTR2. For a set of STRs, SISTR2 takes priors on the parameters of the gamma distribution of fitness effects ($s$), a mutation model, a selection model, demographic parameters, and the observed distribution of heterozygosities across a set of STRs as input. It outputs a posterior estimate of the parameters $a$ and $b$ that describe the gamma DFE across loci ($\mu$: mutation rate; $\Delta$: mutation step size; $x$: STR allele length in number of repeats). Bolded red variables indicate input parameters describing the mutation, selection, and demographic models ($\mu_0$: mutation rate of the optimal allele; $L$: the slope of the increase in $\log_{10}$ mutation rate with allele size; $\beta$: length constraint; $\rho$: step-size parameter; $x_{opt}$: optimal allele; $N$: sample size; $N_{eff}$: effective population size; $n$: number of possible alleles considered at each locus). Bolded blue variables indicate parameters that are estimated by the model ($a$ and $b$ are parameters of the inferred gamma distribution, and mean $s = ab$). The full model is described in **Methods**. (b) Validation of SISTR2 using simulated data. The x-axis indicates the simulated gamma distribution parameters. Dashed vertical lines separate simulation settings with the same mean $s$ value. For a given mean $s$ value, various simulations with different gamma distribution shapes (controlled by $a$) were run. The y-axis gives the estimated mean $s$ of 20 estimates. Black stars indicate the ground truth (simulated) values. (c) Inferred distribution of $s$ values for four gamma distribution parameter ($a$, $b$) combinations. The x-axis denotes bins of $s$ values. The y-axis gives the fraction of loci inferred to be in each bin. The black stars show the ground truth fraction of $s$ values in each bin. For (b) and (c) each color represents a different mutation model setting (**Methods**). Short dinuc = optimal allele 11, long dinuc = optimal allele 20, short trinuc = optimal allele 5, long trinuc = optimal allele 13, short tetranuc = optimal allele 7, long tetranuc = optimal allele 10. (d) Comparison of DFEs inferred by SISTR (per-locus) vs. SISTR2 (joint). Plots show the cumulative distribution of $s$ for various simulation rounds estimated either individually at each locus using SISTR (red) or inferred jointly across all loci using SISTR2 (blue). The black line shows the ground-truth cumulative distribution of $s$ values. The left panel shows a setting with low mutation rate and low selection (mean $s = 10^{-5}$), highlighting a case where SISTR, but not SISTR2, overestimates $s$. The right panel shows a setting with high mutation rate and high selection (mean $s = 0.01$), where both methods perform well. In both settings, SISTR2 produces unbiased estimates of the distribution of $s$ values.

To address these challenges, we developed SISTR2, an extension of SISTR that enables joint estimation of the distribution of fitness effects (DFE) across a set of STRs (**Methods;** Figs 1a and S4). By leveraging information across a set of loci, SISTR2 can obtain more precise estimates of the DFE. Instead of estimating $s$ at each STR individually, the joint method assumes $s$ for each STR is drawn from a gamma distribution, $s \sim \Gamma(a, b)$, and infers the parameters of this distribution.

SISTR2 takes as input allele frequencies for a set of STR loci, prior distributions on the gamma distribution parameters, and the mutation, selection, and demographic models used as input to SISTR. Typically, a set of STRs consists of loci with the same repeat unit and modal (assumed optimal) allele, and SISTR2 is run separately on each set. It first computes heterozygosity (defined as $1 - \Sigma_i p_i^2$, where $p_i$ is the frequency of the $i$th allele) as a summary statistic for each locus in the set. It then uses our simulation framework to simulate a set of STR loci such that the selection coefficient for each locus is drawn from $\Gamma(a, b)$ for some $a$ and $b$. Using ABC, SISTR2 determines the median posterior estimates of $a$ and $b$ by comparing heterozygosity distributions of observed vs. simulated data for different values of these parameters.

Using simulated data, we validated the ability of the joint method implemented in SISTR2 to obtain a posterior estimate of the gamma distribution parameters describing a DFE for a set of STRs. We tested the method on different mutation models as well as a variety of gamma distribution parameters capturing a range of distributions of selective effects. In each simulation, we first chose a mutation model and optimal allele length, then simulated STR allele frequencies for 1,000 loci, drawing the value of $s$ for each locus from a gamma distribution. We then used SISTR2 to estimate $a$ and $b$ from each simulated dataset and compared inferred values to the true values used to simulate the data (**Methods**). We found SISTR2 accurately recovered simulated gamma distribution parameters for mean $s$ values ranging from $10^{-5}$ to $10^{-1}$, that our point estimates of $s$ are unbiased (Fig 1b), and that inferred distributions of $s$ match well to the true distributions used to simulate the data (Fig 1c).

Next, we compared joint estimates of $s$ obtained from SISTR2 to per-locus estimates output by SISTR. To this end, we used SISTR to estimate individual $s$ values for each simulated locus above, and compared the distribution of $s$ values to those from the true underlying gamma distribution (Figs 1d and S5). We found that joint estimation with SISTR2 shows several important advantages. First, it can distinguish between weaker selection (e.g., $s = 10^{-5}$ vs. $10^{-4}$) and neutrality ($s = 0$) which is difficult to do using SISTR. Second, it can more accurately infer selection at loci with low mutation rates. We further evaluated the impact of the number of loci used on SISTR2's performance. As expected, precision increases with the number of loci used as input to the joint estimation. For most settings, accurate estimates of $s$ can be obtained with as few as 10 loci (S6 Fig). For sets of loci with lower mutation rates ($<10^{-5}$), several hundred loci are needed.

Finally, we further evaluated the robustness of SISTR2 inferences to various potential sources of errors. Mean inferred selection coefficients are robust to common STR genotyping errors (S7 Fig) and choice of prior on the gamma distribution parameters (S8 Fig). Results are also not strongly impacted when the optimal allele is not the modal allele if the distribution of true optimal alleles for the set of loci is centered on the assumed optimum (i.e., the modal allele) (S8 Fig). In cases where the observed modal allele is biased to be shorter than the optimum (e.g., as in S2 and S3a-b Figs), assuming the mode is the optimum can result in underestimation of weak selection coefficients. Finally, SISTR2 inferences are sensitive to the choice of mutation model parameters (S8 Fig), motivating our analysis of feasible mutation parameters below before inferring selection coefficients on real data.

## Inferring feasible mutation parameters for various types of STRs using SISTR2

Accurate models of the mutational process are necessary for robust inference of selection on STRs. In addition to inferring DFEs, SISTR2's framework allows us to determine the range of feasible mutation parameters for a set of neutrally evolving loci. For this analysis, we focused on intergenic STRs. Although intergenic STRs could in some cases be pathogenic [10], we assume the majority are under weak or no selection, an assumption that has been made previously for SNPs [11] and supported by the observation that intergenic regions show far less mutational constraint than other genomic regions

[12]. To infer mutation parameters, we set $s = 0$ and considered 6–7 distinct mutation models for each repeat unit length, using SISTR2 to simulate the predicted heterozygosity values. We then compared these distributions of heterozygosity to the empirical distributions, using the proportion of ABC acceptances as a metric of model fit. If multiple mutation models have similar acceptance rates, it indicates that a broad range of mutation models are consistent with the observed heterozygosity distribution or that there is not enough data available to determine which mutation model could explain the observed data. On the other hand, if one model has a higher acceptance rate than others, it indicates that only one mutation model fits best.

We performed STR genotyping using GangSTR [13] in 534 samples of European descent from the 1000 Genomes Project [14] for which deep whole-genome sequencing data was available [15] (**Methods**). We restricted our analysis to STRs with repeat unit lengths 2–4 bp, which are abundant in the genome and can be reliably genotyped. We further filtered very short STRs (**Methods**) since those loci are typically not polymorphic in repeat copy number. After filtering, 86,327 STRs remained for analysis. We then applied SISTR2 using the strategy described above to explore feasible ranges for mutation parameters across STRs with a range of repeat unit lengths, sequences, and modal (optimal) allele lengths. For dinucleotides, we tested 6 different mutation rate models (Fig 2a), and for trinucleotides and tetranucleotides, we tested 7 models (Fig 2b and 2c), each of which models a linear relationship between repeat length and $\log_{10}$ mutation rate. For each setting, we applied SISTR2 to intergenic loci, which we expect to be mostly neutral, and set the gamma distribution parameters $a$ and $b$ such that the $s$ value drawn is always 0. Then, we recorded which mutation models were accepted via ABC for each class of loci and assessed whether accepted simulations were enriched for particular mutation models (Fig 2d-f). The model with the highest number of ABC acceptances corresponds to the best-fitting mutation model.

We assessed the goodness of fit of each model using a post-hoc Kolmogorov-Smirnov (KS) test to compare distributions of heterozygosity values for loci simulated under the model to the observed heterozygosity distribution for a set of STRs (S9 Fig). We repeated this test 100 times for each class of STRs and recorded the percentage of simulation rounds where the observed distributions were significantly different (nominal KS $p < 0.05$). Notably, STR classes with the highest numbers of loci have the highest power to detect even small differences in heterozygosity distributions. To reduce this bias in power, we repeated the analysis after subsetting the number of loci in each class to a maximum of 50 STRs (S10 Fig). For most classes of loci, the percentage of simulations that are similar to the observed values is high, indicating the best-fitting mutation model fits well. However, for certain classes of loci (e.g., long AC repeats, short trinucleotide repeats), simulated and observed heterozygosity distributions are substantially different (KS $p < 0.05$ for many simulations). We plotted examples of observed versus simulated heterozygosity distributions for these cases to visually assess their similarity (S11 Fig). We found in most cases, the observed heterozygosity distribution is still largely similar to that produced by the best-fitting model. Overall, these results suggest that there may be aspects of the mutational process at these loci that are not captured by our model, or that a substantial fraction of loci in those categories are not evolving neutrally. Nevertheless, the best-fitting mutation model still provides a reasonable fit for use in downstream analyses.

We found that even among loci with the same repeat unit length, there is notable variation in feasible mutation rates across different repeat unit sequences. For example, within dinucleotides, we found that a single mutation model fits most loci with repeat units AC and AG well, whereas our analysis suggests most AT repeats have higher mutation rates (Fig 2d). These mutation rates and trends observed here are consistent with mutation rates we [8] and others [16] previously inferred from *de novo* STR mutations, which found that STRs with repeat unit AT mutate several times faster than other dinucleotides STRs (S12 Fig).

Most trinucleotide repeats are fit by a single mutation model, with some notable exceptions (Fig 2e). For example, AAT repeats with modal alleles of 8 or more repeat units are fit best with higher mutation rates than expected even accounting for a linear increase of mutation rate with repeat length. One explanation for this deviation could be that mutation processes at long AAT repeats are not captured by our model and that mutation rates scale super-linearly

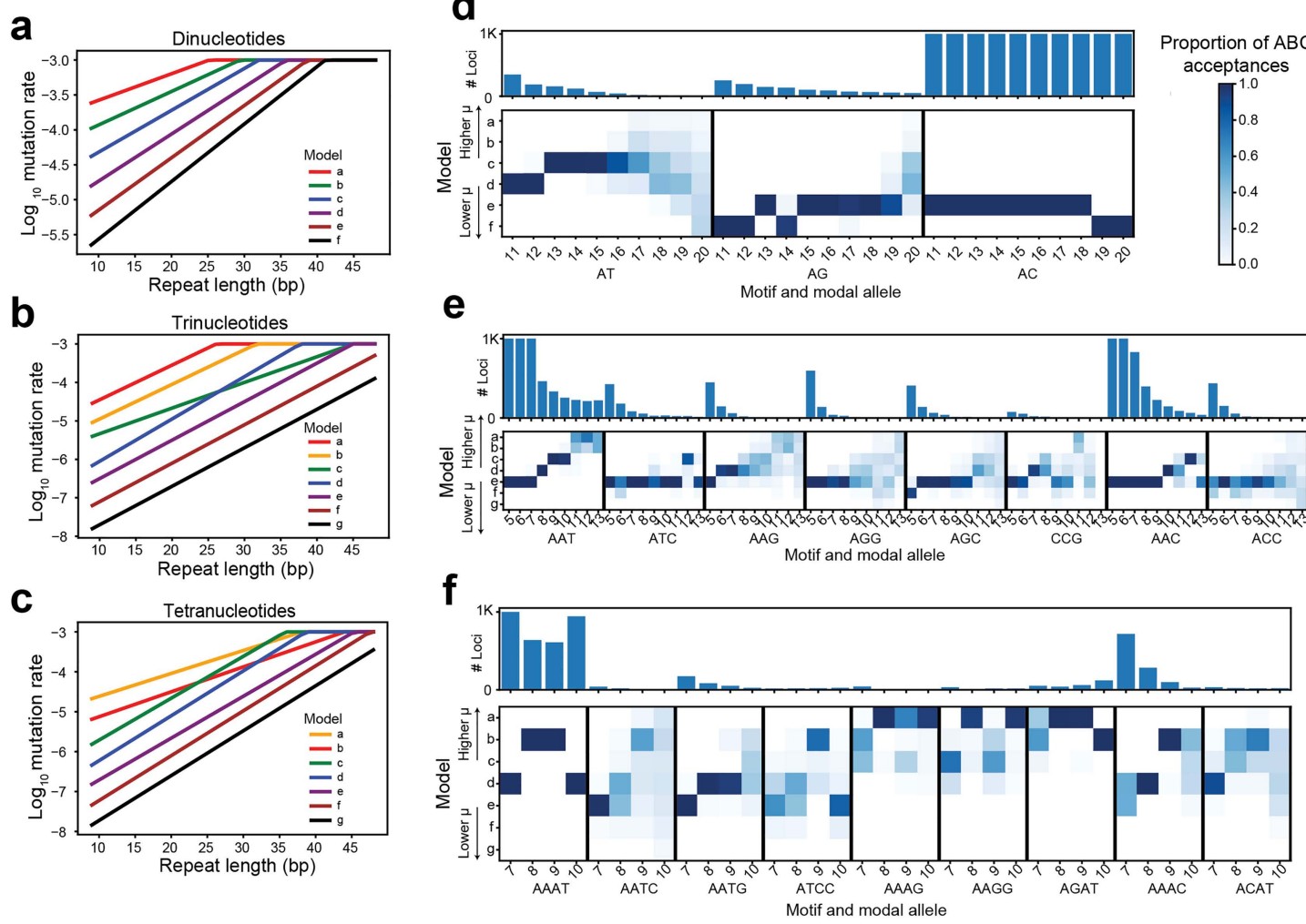

**Fig 2. Exploring feasible mutation rates consistent with observed allele frequencies.** (a-c) Mutation rate models tested for dinucleotide, trinucleotide, and tetranucleotide STRs. In all cases we considered a model with allele-specific mutation rates dependent on the length of each allele. The x-axis shows the allele length, and the y-axis gives the corresponding $\log_{10}$ mutation rate of each allele under each model. In each panel, the different mutation models (a-f for panel a and a-g for panels b and c) all take the form $\log_{10}\mu = \log_{10}\mu_0 + L(x-r)$ where $\mu_0$ is the baseline mutation rate, $r$ is the length of the allele (in number of repeat units) used to define the baseline mutation rate, $L$ gives the linear increase in $\log_{10}$ mutation rate with each additional repeat unit, and $x$ is the length of the current allele (in number of repeat units). Each model uses differing values of $\mu_0$ and $L$. Repeat lengths on the x-axis are denoted in terms of total length in bp. (d-f) Mutation model fit for different STR classes. Each STR class (x-axis) is defined by the repeat sequence and modal (optimal) allele length. Top panels denote the number of loci in each class (truncating at 1,000 for categories with more loci). For bottom panels, each row denotes one mutation setting corresponding to models shown in a-c, with models a-f or a-g generally going from higher to lower overall mutation rates. The color of each cell represents the proportion of the ABC acceptances each mutation model represents.

with repeat length. An alternative explanation is that intergenic AAT repeats may not be truly neutrally evolving, which could bias our mutation model inference. Within trinucleotides, we also estimate that AAG repeats have consistently higher mutation rates, which matches previous observations that these repeats are unstable and prone to large expansions [17].

The best-fitting mutation models for tetranucleotides showed more variability across repeat unit sequences, and are consistent with previous reports that AAAG, AAGG, and AGAT repeats exhibit higher mutation rates [17] and show higher

heterozygosity in human populations [18] than other tetranucleotide STRs. Overall, this analysis highlights substantial differences in mutation rate across STRs with different repeat units. We used these results to inform repeat unit-specific mutation rate parameters for selection inference performed below.

## Distinct types of repeats have different fitness effects

Next, we applied SISTR2 genome-wide to estimate the DFE at different STR classes. As inputs, we used allele frequencies from 86,327 STRs computed from European samples as described above, and set mutation parameters at each class based on the best-fitting mutation model for each repeat unit sequence and optimal allele identified in Fig 2. We first examined DFEs for different repeat classes and functional categories weighted by the number of loci for each optimal allele (Fig 3a, **Methods**). This analysis was restricted to trinucleotide repeats, which were sufficiently abundant across categories. We found, as expected, that coding STRs are under strongest selection whereas intergenic STRs are under the weakest selection (Fig 3a). Intronic STRs and those in UTRs or promoters are under slightly stronger selection than STRs in intergenic regions. Full results for each category are summarized in S1 Table.

Next, we compared DFEs for STRs with different repeat unit sequences (repeat unit length 2–4 bp) and optimal allele lengths (Fig 3b-d and S2 Table). As in our mutation rate analysis, we tested the goodness of fit of the DFE estimated by SISTR2 using a KS-test and found that in most cases, inferred DFEs result in satisfactory fits to observed heterozygosity distributions (S13 Fig). Similar to our mutation rate analysis, observed data at most repeat classes fit well to SISTR2's inferred model with several exceptions. In particular, short trinucleotides, AT repeats, and AAT repeats are not fit well, indicating inferred DFEs for those loci may be unreliable (**Discussion**).

Overall, for dinucleotides, AC repeats tend to be under the strongest selection (Fig 3b). However, this trend is dependent on the length of the STR. Analyzing STRs separately by optimal allele length shows that longer AC repeats tend to be under increased selection, whereas for shorter STRs AG repeats are under strongest selection (S14 Fig). We additionally observed that AT repeats tend to be under weaker selection than AC or AG repeats, although this trend may be driven by the overall shorter average lengths of AT vs. AC repeats (Fig 2) or could reflect the relatively poor fit of our model to AT vs. other dinucleotide repeats (S13 Fig). For trinucleotides, we found that ACC and AAC repeats tend to be under the strongest and weakest selection, respectively (Figs 3c and S14). For tetranucleotides, AAGG, AAAG, and AGAT, which also show higher mutation rates than other tetranucleotide STRs [17] (Fig 2f), exhibit the weakest selection coefficients (Figs 3d and S14).

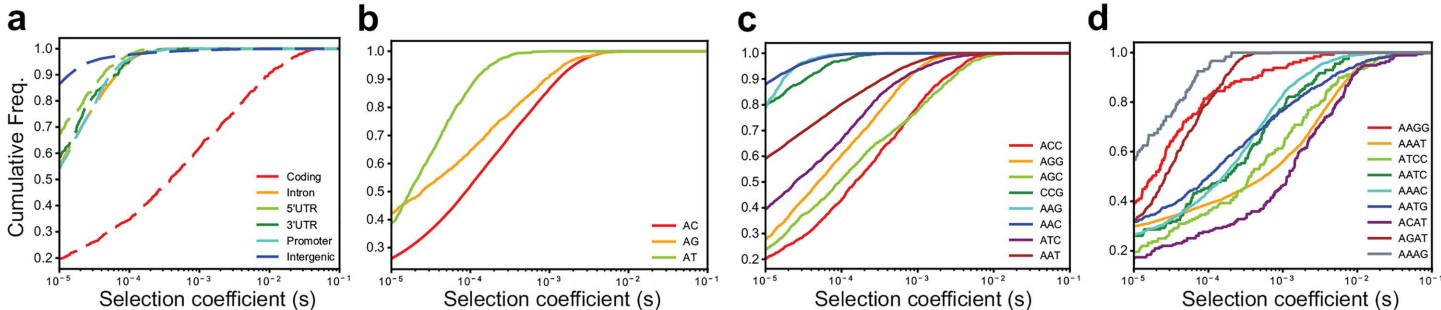

**Fig 3. The distribution of selection coefficients for different STR classes.** (a) Functional category analysis for trinucleotide STRs. Red = coding, orange = introns, light green = 5'UTR, dark green = 3'UTR, cyan = promoters; purple = intergenic. (b-d) Repeat unit analysis. For all plots, the CDF is weighted by the number of loci for each modal allele (**Methods**). Each color denotes a different repeat unit. All analyses presented here use the joint inference method implemented in SISTR2.

## Overall burden of deleterious mutations due to STRs vs. SNVs

We used SISTR2 to estimate the genome-wide fitness burden of *de novo* STR mutations (S3 Table). We assumed the total burden is additive across individual mutations and excluded STR mutation classes which had poor model fit based on the KS-test described above (S15 Fig, **Methods**). This excludes a large number of AAAT and long AC repeats, which likely have deleterious effects but had unreliable estimates for *s*. We computed the predicted burden of mutations within each STR class separately based on their mutation rates, inferred DFEs, and mutation properties, and summed the burden across all classes. Notably, for analyses comparing STR vs. SNV burdens, we present scaled burdens for STRs, in which *s* values are divided by 2, to match how SNV selection coefficients were previously defined (see **Methods**).

We next compared the estimated burden of genome-wide *de novo* STR vs. SNV mutations (Table 1). For SNVs, we considered two classes of mutations for which mutation rates and fitness effects were available [11,19]: nonsynonymous mutations and mutations occurring in conserved noncoding regions (**Methods,** S4 Table). The expected fitness effect of an individual single-nucleotide nonsynonymous mutation ($s = 6.6 \times 10^{-3}$) is approximately 17 times higher than that of an individual STR mutation (scaled $s = 3.8 \times 10^{-4}$), whereas the effect of an individual single-nucleotide mutation in conserved noncoding regions ($s = 5.6 \times 10^{-4}$) is similar to that of an STR mutation. However, each individual genome is expected to have far more *de novo* STR mutations (mean $= 14$, considering genome-wide mutations at STRs with repeat units 2–4 bp and reliable selection models) compared to mutations resulting in nonsynonymous SNVs (mean $= 0.56$) or SNVs in conserved noncoding regions (mean $= 2.5$). Thus overall, the expected total burden of genome-wide STR mutations is notably higher than either category of single-nucleotide mutations (approximately 1.7x and 4.4x higher than nonsynonymous and conserved noncoding, respectively) and is higher than both categories of single-nucleotide mutations together.

The analysis above may overestimate the relative burden of STR vs. SNV mutations since it considers genome-wide STRs but only a subset of possible SNV mutations collectively covering less than 5% of the genome. We considered the theoretical mutation burden based on varying the proportion of noncoding single-nucleotide mutations under selection from 5% to 30% (S4 Table), and found that the overall burden from noncoding SNVs when up to 20% of the noncoding genome is under selection (0.00571) is still smaller than genome-wide scaled STR burden (0.0062). Only when 30% of the noncoding genome is affected by negative selection does the burden from noncoding SNVs (0.00857) surpass that from STRs. Overall, while this analysis faces several limitations (**Discussion**), our results suggest that although STR mutations individually have modest fitness effects, the larger number of STR mutations per generation results in a total fitness burden that is several times higher than that of nonsynonymous, and similar or higher in magnitude to the genome-wide burden of all single-nucleotide mutations.

Finally, we compared the relative burden of inherited STRs vs. SNVs in a typical genome (S5 and S6 Tables). For STRs, we computed the fitness effects of inherited variation in the children of three trios in the 1000 Genomes dataset,

**Table 1. Comparison of estimated fitness burden from *de novo* STR vs. SNV mutations. Burden refers to the sum of fitness effects across all mutations in each category. SNV burdens are computed based on average mutation rates and fitness effects for each class (Methods). For STRs, the expected burden is summed across the burden from each STR class (repeat unit and optimal allele). 95% confidence intervals were computed based on 100 bootstrap samples of mutations sampled from each class based on their relative proportions and sampling their selection coefficients from the gamma distribution inferred by SISTR2 (Methods).**

| Mutation type | Mean $s^*$ | # Mutations/generation | Total burden* |
|---|---|---|---|
| STRs - genome-wide+ | 0.00038 | 13.78 | 0.0062 (0.00040-0.020) |
| SNVs - nonsynonymous | 0.0066 | 0.56 | 0.0037 |
| SNVs - conserved noncoding | 0.00056 | 2.54 | 0.0014 |

+ STR counts only consider repeats with units 2–4 bp that have repeat units common enough to be estimated by SISTR2 and with relatively good model fits (S2 Table).

*To enable direct comparison with SNV burden, for which *s* refers to selection on heterozygous genotypes, STR burden values, for which *s* was defined to refer to selection on individuals homozygous for a particular allele, are divided by 2 compared to values shown in S3 Table (see **Methods**).

excluding repeat classes with poor model fit (**Methods**). For comparison, we computed effects of noninherited variants in each trio, which may arise either due to *de novo* mutations or cell line artifacts. As expected, inherited STR variants tended to have lower selection coefficients compared to these noninherited variants (S16 Fig; KS two-sided p < 0.01 for all samples). For SNVs, we considered observed variants in two classes: nonsynonymous variants and variants falling in conserved noncoding regions as measured by CADD [20] score >15, corresponding to the top 3% of genome-wide scores (**Methods**). The estimated STR burden is substantially lower than that for SNVs in these categories (mean scaled STR burden 4.41 excluding long AC/AAAT repeats, and mean SNV burden 27). The majority of this SNV burden is attributed to nonsynonymous SNVs (20 vs. 6.8 for conserved noncoding). Similar to our estimates for *de novo* mutations, we additionally varied the proportion of noncoding variants potentially under selection by considering a range of CADD score thresholds. For more permissive definitions of conserved noncoding regions (CADD>10 or >5), the SNV burden increased to 44 and 127, respectively. Overall, our results suggest that while the total burden of *de novo* mutations is larger for STRs, the burden of inherited variants is larger for SNVs, regardless of how conserved noncoding regions were defined.

## Discussion

Here, we presented SISTR2, a method for joint inference of the distribution of fitness effects (DFE) across a set of STRs. SISTR2 allows for improved inference of selection compared to SISTR at a broad range of repeat classes including those with low mutation rates or under only weak negative selection. We additionally leverage SISTR2 to refine estimates of STR mutation parameters, infer selection parameters across a diverse set of repeats, and estimate the relative burden of STR vs. SNV mutations in a typical genome.

We found that mutation and selection parameters are highly variable across STR classes. For example, we found that AT, AAG, AAAG, AAGG, and AGAT repeats have notably higher mutation rates than other STRs with the same repeat unit length. Further, while mutation rates for the majority of STRs scale linearly with repeat unit length, we found evidence that AAT repeats scale super-linearly with repeat length. Similarly, we estimate that STRs exhibit a wide range of selection coefficients, depending on the repeat unit and functional annotation of the repeat (Fig 3). This wide variation in mutation and selection parameters across STRs is consistent with the complex fitness landscapes recently reported in a more detailed study of selection at 94 STRs [6].

We used SISTR2 to compare the expected overall fitness burden per individual of STR vs. SNV mutations. We considered both the burden of *de novo* mutations, as well as the burden due to inherited variants across the genome. The expected fitness of an individual STR mutation is estimated to be modest (17-fold less than a nonsynonymous SNV). However, STRs are highly prevalent and experience a far greater number of mutations per individual. Thus, we estimate the *de novo* burden of STR mutations to be greater than that of SNVs. On the other hand, we found that the total burden of inherited variation is likely larger for SNVs compared to STRs. We hypothesize that this may be because the total space of possible SNV mutations is much larger than that for STRs. Whereas only a small number of SNVs occur per generation, they accumulate over time as more sites are mutated. In contrast, STR mutations largely occur at a predetermined set of repeat elements already present in the genome. Further, STRs experience frequent "back" mutations which may reverse the effects of deleterious mutations in previous generations. Notably, our analysis was restricted to a subset of STR mutations with repeat unit lengths 2–4 bp that could be analyzed by SISTR2, and is thus likely an underestimate of the total burden of STR mutations. Still, our inferences of relative burden for both *de novo* and inherited variants are robust across a range of definitions of the sets of SNVs and STRs used for analysis.

Our study faced several limitations. First, one of the most significant challenges of our model is that we assume a single known optimal allele length at each locus and must analyze STRs with different optimal alleles separately. In practice, this optimum is unknown and is set to the modal allele. Encouragingly, a recent study inferred the optimal allele through population genetic models and found that it often corresponded to the modal allele in the population when there was a

single optimal allele [6]. In agreement with this, we found that for the mutation and selection models here, STRs that are under at least weak selection almost always have modal alleles that match the optimum (S1 and S2 Figs). As expected, this assumption is not true at loci that have both very weak selection and rapid mutation rates, which would have been unlikely to be detected as being under selection regardless of how the optimal allele is identified. The modal allele also may not correspond to the optimal allele at loci with long optimal allele lengths, where the length-dependent mutation rate can result in an accumulation of alleles that are shorter than the true optimum since those alleles exhibit lower mutation rates. Nevertheless, we found that our model is relatively robust to errors in inferring the optimal allele (S8 Fig). Extending our model to relax this assumption or consider alternative models such as directional, rather than symmetric, selection models is a topic of future study.

Second, while our post-hoc analysis suggests inferred mutation and selection models fit the majority of STR classes well (S9-S10 and S13 Figs), for some classes such as short trinucleotides or long AC and AAT repeats, we could not identify a single best-fit model. This suggests for some repeat classes that our models do not completely capture properties of STR mutation, such as the influence of flanking sequence features [18] or sequence imperfections [21], potentially biasing estimates of selection. Third, the model of selection within SISTR and SISTR2 assumes that mutations away from the modal allele are deleterious and that mutations act in an additive manner. In reality, fitness landscapes for STRs may be more complex. A recent study found that some STRs either may exhibit dominance effects or periodic fitness surfaces, with multiple optimal alleles [6]. Future work could examine whether more elaborate models, including positive selection and dominance effects might better fit the data. Finally, results here are based on allele frequencies from individuals of European descent. Computation of DFEs, and comparison of per-locus STR selection coefficients across populations, is an important topic of future work.

Overall, our findings suggest STRs are widespread targets of natural selection and that mutations at STRs contribute a substantial fitness burden in humans. Further, our results highlight important differences in mutation and selection across STR classes. These findings will inform future methods by enabling more accurate modeling of mutation and selection processes at STRs including improved inference of the impact of individual mutations that may contribute to evolution and disease risk.

## Methods

### SISTR2 mutation, selection, and demographic models

The mutation, selection, and demographic history models are the same as those used previously in SISTR [8]. The models and rationale for model choices are briefly summarized here for clarity.

**Mutation model.** Our mutation model has four input parameters ($\mu_0$, $L$, $\rho$, and $\beta$) which are described below. We use a mutation model based on the generalized stepwise mutation model (GSM) [22], which considers STRs to mutate with some probability $\mu$ each generation and allows mutations consisting of one or more repeat units. We make several modifications to the classical model:

First, based on the widespread finding that STR mutation rate is strongly correlated with the total repeat length [7,8,16,23], we apply a length-dependent mutation rate. The mutation rate $\mu_x$ of an allele with size $x$ repeat copies is determined by $\log_{10}(\mu_x) = \log_{10}(\mu_0) + Lx$, where $\mu_0$ is the mutation rate of the optimal allele ($x_{opt}$) and $L$ is the length-dependent mutation rate.

Second, based on observed step sizes from *de novo* mutation data [8,16,24–26], we model the size of each mutation (number of repeat units inserted or deleted) based on a geometric distribution with parameter $\rho$. Under this model, single-step mutations are most common (with probability $\rho$), but multi-step mutations are allowed.

Third, whereas the traditional GSM assumes insertion or deletion mutations are equally likely, we incorporate a direction bias, in which long alleles are more likely to contract and short alleles are more likely to expand toward the optimal allele length. This choice is similarly based on multiple previous studies that have observed a length-dependent bias in

mutation direction either directly from *de novo* mutations [8,16,24,27] or indirectly through patterns of variation at STRs across populations [7,28]. Let $k = x - x_{opt}$ be the size of an allele relative to the optimal allele. Under our model, we set the probability that an allele of size $k$ will expand to $\frac{1-\beta\rho k}{2}$ and the probability it will contract to $\frac{1+\beta\rho k}{2}$, where $\beta$ is the length constraint and $\rho$ is the step size parameter described above.

**Selection model.** The selection model assumes an optimal allele with fitness 1, with the fitness of other alleles decreasing with the number of units away from the optimal allele. More formally, the fitness ($w$) of each allele $x$ is given by $w(x) = 1 - s|x - x_{opt}|$, where $s$ is the selection coefficient. We assume an additive fitness model, where the fitness of an individual with diploid genotype $(x_1, x_2)$ is defined as the average of the fitness of their two alleles at a locus ($\frac{w(x_1) + w(x_2)}{2}$). As such, under this definition, SISTR2 defines $s$ to be the reduction in fitness for an individual who is homozygous for the deleterious allele and $0.5s$ would reflect the fitness effect of a single copy of the allele. Unless otherwise noted, $s$ for SISTR2 results reflects the homozygous fitness effect.

**Demographic model.** SISTR uses a European demographic model with previously published parameters [29,30]. The model starts with an ancestral population of size 7,310. After 50,000 generations, the population goes through an expansion 5,920 generations ago to a population size of 14,474; an out of Africa bottleneck event 2,040 generations ago to a population size of 1,861; and a European/Asian divergence event 920 generations ago to a population size of 1,032. These events are then followed by two periods of recent exponential growth. From 920 to 205 generations ago, the population grows at a rate of 0.00307; from 205 generations ago to present, the growth rate increases to 0.0195.

## Forward simulation of allele frequencies

Forward simulation is performed using the algorithm described for SISTR [8] based on the method introduced by Haasl and Payesur [5]. Briefly, we assume there are $n$ possible alleles at each locus (in all cases we set $n = 25$). We simulate a vector of allele frequencies for each locus. The vector is initialized such that the optimal allele has frequency 1 and all other alleles have frequency 0. For each generation, we recalculate allele frequencies after one generation of mutation and selection. We then simulate drift by performing multinomial sampling to draw a sample of size $2N_e$, where $N_e$ is the effective population size, and use sampled alleles to recalculate allele frequencies. $N_e$ is based on the demographic model used and can vary across generations. This process is repeated for the specified number of generations based on the demographic model (here, 50,000 initial generations at a constant effective population size followed by 5,920 generations of size changes described above). After the final generation, an end sampling step is performed, in which we recompute allele frequencies based on multinomial sampling to draw a sample of size $N$, where $N$ is the number of alleles observed in empirical allele frequency data.

## Estimating the distribution of selection coefficients using SISTR2

SISTR2 uses ABC to estimate a distribution of $s$ values (the distribution of fitness effects, or DFE) across a set of STRs (Fig 1a). Here, sets of STRs are typically defined as STR loci with the same repeat unit sequence (e.g., AAAT repeats) and modal allele, which is assumed to match the optimal allele (see **Discussion**). SISTR2 is run separately on each of these sets defined by their repeat unit/modal allele. For each set, we assume the value of $s_i$ at each STR is drawn from $\Gamma(a,b)$ and learn posterior distributions for $a$ and $b$. SISTR2 takes as input summary statistics computed from allele frequencies for each of the set of STRs, plus a mutation model, selection model, demographic history model, and a prior distribution on the gamma distribution parameters $a$ and $b$. It then outputs a posterior estimate of the gamma distribution parameters.

We use heterozygosity as a summary statistic to describe each STR locus. The heterozygosity of locus $i$ is defined as $h_i = 1 - \sum_j p_{ij}^2$, where $p_{ij}$ gives the frequency of the $j$th allele at locus $i$. Notably, unlike most SNPs, STRs often have more than two unique alleles at each locus. STRs for which a certain allele is fixed in the population have heterozygosity

0, whereas heterozygosity approaches 1 for highly variable STRs. We used this summary statistic since it captures additional information about the distribution of alleles at each locus beyond alternative choices considered (including frequency of the most common allele or number of common alleles, both of which only capture information about a subset of alleles).

To reduce computational complexity, SISTR2 first draws a random subset of 1,000 loci from the input set of STRs (or uses all the loci if there are less than 1,000) and obtains the heterozygosity for each STR. It then repeats the following steps $z$ (typically 50,000) times:

1. Draw gamma distribution parameters $(a, b)$ from the priors, where $a$ is drawn from a uniform distribution from 0 to 1 and $a * b$ (mean value of the distribution) is drawn from a log normal distribution where the underlying normal distribution has a mean of log(0.0003) and a standard deviation of log(30). These parameters capture the range of possible selective effects and are guided by previous work on SNVs. Previous estimates of the shape parameter of a gamma DFE are consistently leptokurtic ($a < 1$) for SNVs [11,19] and estimates of the mean strength of selection are ~0.00065-0.0066 for noncoding and nonsynonymous SNVs [11,19] (Table 1), which are within our prior on $a * b$. We found that SISTR2 estimates are robust to modifications of the exact constants used for the mean and standard deviation of the log normal distribution (S8 Fig).

2. For each STR $i$ in the subset, draw $s_i$ from the gamma distribution, simulate allele frequencies forward in time using this value of $s_i$, and compute the heterozygosity of the resulting allele frequencies. Forward simulation is performed using a previously described algorithm from Haasl and Payseur [5] as implemented in SISTR [8] that simulates allele frequencies for $n$ possible alleles (typically $n$=25) at each locus. The simulation assumes an effective diploid population size ($N_{eff}$) which varies across generations according to a published model of European population history [30] and includes an end sampling step based on the sample size ($N$) of the empirical dataset.

3. Compare the distribution of heterozygosity values for all loci within the simulated set to the distribution of observed heterozygosity across input loci. For this, we sort vectors of heterozygosity values separately for the simulated and observed loci. We then summarize the difference in distributions by obtaining the difference of these two vectors and taking the mean of the absolute value of the differences. Notably, this metric could be impacted by loci with extreme heterozygosity values. In practice we found that using this metric achieved good model fit at most STR classes and that cases of poorer fit were not driven by outlier heterozygosity values (S11 Fig).

The $a,b$ parameters that generate the top 1% of simulated heterozygosity distributions that are most similar to the observed distribution based on the mean of differences in step 3 above are accepted. We report the posterior estimate of the gamma distribution as the $a, b$ pair with the median value of $a*b$ (mean selection coefficient) out of all the accepted pairs. In practice to improve computational efficiency, we first generate a lookup table for each STR class (determined by its period and optimal allele) containing a list of selection coefficients and corresponding allele frequency summary statistics. Then, for step 2 above, instead of simulating allele frequencies for each STR, we determine what class it belongs to and use the corresponding lookup table and randomly select a value of $s_i$ sufficiently close to the $s_i$ value drawn from the gamma distribution. For this, we rounded each $s$ value to a single significant digit. Values <$10^{-5}$ were rounded to 0. Two $s$ values were considered sufficiently close if their rounded values matched.

## Validation of SISTR2 using simulated data

We validated SISTR2 using simulated datasets for six classes of STRs defined by their period (length in bp of the repeat unit) and optimal allele length. For each class we analyzed 12 pairs of gamma distribution parameters $(a, b)$. For each class of STRs, for each gamma distribution parameter pair $(a, b)$, we obtained the ground truth heterozygosity distribution obtained from simulating 1000 allele distributions using the mutation model for that class and $s$ values drawn from the

gamma distribution characterized by $s \sim \Gamma(a, b)$. Then, we performed ABC 20 times (each time with 2,000 simulations) to obtain 20 posterior estimates of the gamma distribution parameters as described above. Next, we calculated the mean $s$ value (given by $a*b$) of each posterior estimate of $a,b$ and plotted the mean of the 20 mean $s$ values of estimated $a,b$ parameters (Fig 1b). To obtain the distribution of $s$ values in different $s$ bins for a given pair of parameters ($a,b$), we drew 1,000 $s$ values using ($a,b$) and calculated the fraction of $s$ values in each bin (Fig 1c).

### Simulating genotyping errors

To evaluate the impact of STR genotyping errors on the results of SISTR2, we modified our simulation framework to add errors to simulated observed genotypes. We set the error probability for each observed allele to 0.1%. Each incorrect allele was set with 50% probability to be either one repeat larger or shorter than the true allele length. We then re-ran SISTR2 with these noisy observed genotypes as input (S7 Fig).

### Genotyping STRs in European samples from the 1000 Genomes Project

Aligned whole genome sequencing CRAM files for the European (except Finnish) samples of the 1000 Genomes Project were obtained from SRA accessions PRJEB31736 (unrelated samples) and PRJEB36890 (related samples). We excluded Finnish samples due to their unique population demographic history [14].

GangSTR [13] v2.4.5 was run on each sample separately with nondefault parameters --str-info str_info_file (see below), --bam-samps sample_id, --samp-sex sample_sex, and --grid-threshold 250. We generated an initial set of reference STRs for the hg38 assembly using Tandem Repeats Finder [31] with the following parameters: match = 2, mismatch = 5, indel = 17, maxperiod = 20, pm = 80, pi = 10 and minscore = 24. We then refined the STR reference set by applying a series of filtering steps. First, we removed repeats longer than 1kb. Then, we kept a single repeat with the shortest motif length among those with identical start or stop coordinates. Compound and imperfect repeats were removed and any extra bases not matching the repeat motif were trimmed from both sides. Any duplicated repeats were discarded post-trimming. Finally, we filtered out any overlapping repeats if their motifs consisted of identical nucleotide types.

The file str_info_file contains the per-locus stutter parameters obtained by training the stutter model on 19 samples using a modified version of HipSTR v0.6.2 (https://github.com/mikmaksi/HipSTR) with nondefault parameters --stutter-model-only (to skip genotyping), --chrom (to run separately for each chromosome), --min-reads 20, and --output-filters.

In the first merging step, mergeSTR [32] v3.0.3 with nondefault parameter --vcftype gangstr was used to merge the VCF files of each sample into a unified VCF file for each sub-population (CEU, TSI, GBR, and IBS). The second merging step also uses mergeSTR v3.0.3 with nondefault parameter --vcftype gangstr to merge all of the sub-population level VCFs into one unified merged VCF for all European (except Finnish) samples. This merged VCF file was then filtered using dumpSTR [32] v3.0.3 using nondefault parameters --vcftype gangstr, --min-locus-callrate 0.8, --min-locus-hwep 0.00001, --filter-regions SEGDUP.bed, --filter-regions-names SEGDUP, --gangstr-filter-spanbound-only, --gangstr-filter-badCI, --gangstr-min-call-DP 20, --gangstr-max-call-DP 1000, --gangstr-require-support 2, and --gangstr-readlen 150. A list of segmental duplications (SEGDUP.bed) for the hg38 reference genome build was obtained from UCSC table browser [33].

We then used statstr [32] v3.0.3 with options --acount --numcalled to compute per-locus allele counts and record the number of genotyped samples per locus. We then filtered STRs with call rates <80%. We further excluded TRs with repeat lengths in hg38 < 11 units for dinucleotides, < 5 units for trinucleotides, and <7 repeats for tetranucleotides, since those repeats are typically not polymorphic. After filtering 86,327 STRs remained for analysis. The genomic annotation of each STR was assigned based on Ensembl [34] build 92 for the GRCh38 reference genome. Promoters were defined as regions within 5kb upstream of transcription start sites of protein coding genes.

## Estimating feasible mutation rates

For each type of STR (e.g., dinucleotide with a particular repeat unit and modal allele), we examined 6–7 different mutation models. All models take the form $\log_{10}\mu = \log_{10}\mu_0 + L(x-r)$, where $\mu_0$ is the baseline mutation rate, $r$ is the length of the allele (in number of repeats) used to define the baseline mutation rate, $L$ is the linear increase in $\log_{10}$ mutation rate with each additional repeat unit, and $x$ is the length of the current allele (in number of repeat units). We chose a range of values for $L$ and $\mu_0$ based on empirically observed trends in *de novo* mutation data [8,16]. We set the maximum possible mutation rate to $10^{-3}$ and the minimum possible rate to $10^{-8}$. For each mutation model, we set $s=0$ and used the proportion of acceptances in the ABC framework as a measure of model fit. With a uniform prior on the mutation rate models, the proportion of acceptances approximates the likelihood of the model, given the data [35–37], and thus the best-fit model can be interpreted as the maximum likelihood model.

## Computing weighted CDFs of selection coefficients

Cumulative distribution function plots (Fig 3) were computed by weighting results across DFEs inferred for all optimal allele lengths for each STR class. For each class, for each optimal allele we drew a number of $s$ values from the learned DFE equal to the number of loci in that class. We then combined these randomly sampled $s$ values across models for all optimal alleles, and computed cumulative distributions based on these combined values.

## Computing the expected burden of *de novo* STR mutations

For each STR class $c$ (motif/optimal allele length category), we used the model
$Burden_c = N_c * 2 * \mu_c * \sum_{i=-20...20} (P(mutation_i) * s_c * 0.5 * |i|)$ to compute the expected burden of *de novo* mutations for that class of STRs. $N_c$ gives the number of loci in class $c$; the factor of 2 denotes the fact that humans are diploid and each homologous chromosome can mutate; $\mu_c$ gives the mutation rate of the optimal (modal) allele for class $c$; the factor of 0.5 converts the selection coefficient from SISTR2 (defined as the strength of selection for homozygotes for the given allele) to the selection strength for a single copy of the allele; $i$ represents all possible mutation lengths and ranges from -20–20 repeat units since mutations of larger sizes are extremely rare; $P(mutation_i)$ is the probability of a mutation resulting in a change of $i$ repeat copies, and is computed based on the step size parameter $\rho$ (see mutation model described above). $s_c$ represents the mean selection coefficient for an STR in class $c$ computed by SISTR2 (S2 Table). The expected burdens from all the classes were then summed to obtain the genome-wide burden (S3 Table). We excluded STR classes with poor fit (KS Score$\leq$50, S2 Table) from downstream analyses.

## Computing the burden of inherited STR variants

We computed the burden of inherited STR variants using children from three trios (samples NA12864, NA10865, and NA10845 from the 1000 Genomes Project [14]. The DFE for *de novo* mutations differs from the DFE for standing variation since the DFE for standing variation is biased toward more neutral values of $s$. For example, the $s$ values for loci with inherited mutations that are a large number of repeat units away from the optimal alleles are likely not a random draw of $s$ from the DFE inferred from SISTR2. Instead, these variants likely occur at loci that have values of $s$ that are from the neutral portion of the DFE, otherwise those variants would have been removed by negative selection. Therefore, to compute the burden of inherited STR variants, we took a Bayesian approach that takes into account the frequency of a particular allele that is segregating in the population in the present-day.

For a given individual, for each STR variant at a locus with a SISTR2 score, we found which class $c$ (motif/optimal allele category) the STR belongs to. Then, we calculated the reduction in fitness as $|allele - optimal\ allele| * s$. To obtain the $s$ for each inherited allele, 1,000 values of $s$ from the gamma distribution corresponding to class $c$ were drawn and allele frequencies were simulated using those values of $s$. For each of the 1,000 $s$ values, if the simulated

allele frequencies include the observed STR allele, the value of *s* and the STR allele's frequency were recorded. For example, if we drew *s* = 0.001 and it gave the observed STR variant (15) and the allele 15 has a frequency of 0.2, we would record the pair (0.001, 0.2). From this, we generate a table of (*s* value, frequency of desired STR allele) pairs. We then draw an *s* value from this table, with probability equal to the frequency of the mutant allele. For instance, if the table has (0.0001, 0.5), (0.001, 0.2), (0.01, 0.1), we would draw 0.0001 with a 5/8 chance, 0.001 with a 1/4 chance, and 0.01 with a 1/8 chance.

Burdens less than $10^{-5}$ were rounded down to 0 and greater than 1 were set to 1. Furthermore, if a variant was not found in any of the 1000 simulations, *s* was set to 0. Finally, for variants that were greater than 12 repeat units away from the optimal allele, *s* was automatically set to 0 since the simulations only contain 25 alleles in total. The fitness reductions for all STRs were summed to obtain the total STR burden for the individual. Mutations at STR classes with poor model fit (defined above) were excluded from analysis.

**Identifying noninherited STR variants**

We additionally computed the observed burden of noninherited mutations in the three trios analyzed above. Because the data is derived from lymphoblastoid cell lines (LCLs), noninherited mutations may correspond either to *de novo* mutations or cell line artifacts such as mosaicism [23,38]. We first genotyped STRs using GangSTR as described above. We then used mergeSTR from the TRTools toolkit v4.0.0 to merge VCF files of individuals in each subpopulation. We applied call-level filtering on autosomal chromosomes of resulted VCF file using dumpSTR, from the TRTools toolkit v4.0.0 with parameters --min-call-DP 20, --max-call-DP 1000, --filter-spanbound-only, and --filter-badCI.

Then, filtered genotypes were subject to locus-level filtering with the parameters --filter-regions hg38_segdup_sorted. bed.gz, --filter-regions-names SEGDUP, --min-locus-callrate 0.8, and --drop-filtered to remove those loci that overlap with segmental duplications and those with low call rate. We then used statSTR from TRTools v4.0.0 to check consistency of parent genotypes with Hardy Weinberg Equilibrium (HWE) and used bcftools v1.12.14 to filter loci with a HWE p-value less than $10^{-5}$.

Finally, MonSTR [8] v2.0 was used to call *de novo* mutations with nondefault parameters --naive, --gangstr, --min-num-encl-child 3, --max-perc-encl-parent 0.05, --min-encl-match 0.9, --min-total-encl 10. We further removed *de novo* calls that were homozygous, removed loci that were biased toward expansion and deletion (two-sided binomial p-value < 0.05), removed mutations for which the *de novo* allele was supported by any reads in the parents and fewer than 5 reads in the child, and removed STRs for which more than 6 mutations across 568 total 1000 Genomes trios were observed implying that they are most likely error-prone STRs.

To compute the burden of each mutation, we scored the fitness of each allele as |*allele-optimal allele*|**s*, where *s* is the mean selection coefficient for that STR according to its class (motif/optimal allele category) based on SISTR2. Mutations at STR classes with poor model fit were excluded from analysis. The fitness effect of each mutation was computed as the difference in fitness between the new allele and the parent allele.

Note that for all comparisons between STR and SNV burdens, we divide *s* for STRs by 2 so that it refers to the effect of an allele and is comparable to how *s* is defined for SNVs (see below).

**Computing the burden of *de novo* single nucleotide mutations**

To compute the burden of SNV mutations, we turned to previous estimates of the DFE for nonsynonymous mutations [19] as well as for conserved noncoding mutations [11]. The mutational burden for each category was calculated as $Burden = 2 * L * \mu * s$, where *L* represents the mutational target size (i.e., number of sites that could be mutated) in a haploid genome, $\mu$ the per-base pair mutation rate, and *s*, the mean selection coefficient (where *s* is the selection strength on a heterozygote and 2*s* is the strength of selection on the homozygote), found from the inferred DFE (S4 Table). Mutational target sizes for noncoding regions were taken from Huber, *et al*. [39]. Given that the number of noncoding sites

where mutations could have fitness effects is not precisely known, we explored a range of values ranging from 5% to 30% of noncoding sites being under selection (S4 Table).

## Computing the burden of inherited SNV variation

To compute the burden from standing genetic variation, we used an approach that links an estimate of the DFE to standing genetic variation via forward simulations. First, we used SLiM [40] to simulate nonsynonymous variation in a European population using a demographic history from Gravel *et al.* 2011 [29] and a DFE from Kim *et al.* 2017 [19]. Similarly, we simulated noncoding variation using a DFE from Torgerson et al 2009 [11]. From these simulations, we obtained the distribution of selective effects for heterozygous nonsynonymous, homozygous nonsynonymous, heterozygous conserved noncoding, and homozygous noncoding variants. Then, using data from the 1000 Genomes high coverage sequencing dataset [15], we computed the number of heterozygous nonsynonymous, homozygous nonsynonymous, heterozygous conserved noncoding, and homozygous conserved noncoding variants, averaging over six individuals (NA06985, NA06986, NA06994, HG02234, HG02237, HG02240). In the real data, conserved noncoding mutations were annotated as having a CADD score >15 and annotations for nonsynonymous mutations were obtained from files released by the 1000 Genomes project team (https://www.internationalgenome.org/data-portal/data-collection/30x-grch38).

For each mutation in each individual, we drew a selection coefficient from the simulated distribution of standing genetic variation, conditional on whether the mutation was conserved noncoding or nonsynonymous and whether it was observed in a heterozygous or homozygous state. We computed additive fitness as $\sum_{i=1}^{M} s_i^{het} + 2 * \sum_{j=1}^{N} s_j^{hom}$ separately for both nonsynonymous and conserved noncoding, where $M$ is the number of heterozygous sites in an individual and $N$ is the number of homozygous sites in an individual. Note, for this analysis, both $s_i^{het}$ and $s_i^{hom}$ represent the strength of selection a segregating heterozygote. This necessitates the factor of 2 in the additive fitness equation for the homozygous genotypes such that $2 * s_i^{hom}$ is the strength of selection on a homozygote. Note that for comparisons between STRs and SNVs, we divide $s$ for STRs by 2 so that it refers to the effect of an allele and is comparable to how $s$ is defined for SNVs.

## Supporting information

**S1 Table. SISTR2 scores by functional annotation.** SISTR2 was used to infer the distribution of selection coefficients at different classes of trinucleotide STRs overlapping different genomic annotations. Columns give the repeat unit length (in bp), optimal allele length (in rpt. units), category (coding, intron, 5'UTR, 3'UTR, promoter, intergenic), and number of loci considered for each inference. The last three columns show the estimated values for the gamma distribution parameters $a$ and $b$ as well as the corresponding mean estimated selection coefficient (computed as $a*b$).
(XLSX)

**S2 Table. SISTR2 scores by repeat unit/opt allele.** SISTR2 was used to infer the distribution of selection coefficients at sets of STRs with different repeat unit sequences and optimal allele lengths. Columns give the optimal allele length (in rpt. units), repeat unit sequence, and number of loci in each class used for inference. The last three columns show the estimated values for the gamma distribution parameters $a$ and $b$ as well as the corresponding mean estimated selection coefficient (computed as $a*b$). The KS Score is based on comparing simulated heterozygosities for loci under our inferred model to observed heterozygosities using a Kolmogorov-Smirnov test. The score gives the number of simulations (out of 100) for which the KS test returned $p$-value > 0.05, indicating a good fit.
(XLSX)

**S3 Table. Genome-wide burden of STR mutations.** For each class of STR (repeat unit sequence/optimal [modal] allele combination), the table shows the number of loci, mean burden of each new mutation computed as the selection coefficient for that class times the mean mutation size, the scaled mean burden of each new mutation (where the original value is divided by 2), mutation rate for that class, and total estimated burden contributed by each class. Burden is computed as

2*mutation rate*mean scaled burden*number of loci. The KS Score is the same one presented in S2 Table and is used to filter out loci for the burden analysis where the mutational model did not fit well. Only loci with scores >50 were retained. The number of mutations/category reflects 2*mutation rate*number of loci in the category. The column "mean burden/ mutation (rescaled) x number of loci" reflects the mean rescaled burden multiplied by the number of loci, which is then used to estimate the mean rescaled *s* shown in Table 1.
(XLSX)

**S4 Table. Fitness effects of *de novo* SNVs.** Columns give the SNV mutation type (nonsynonymous, conserved noncoding, or noncoding considering various percentages of noncoding sites under selection), mean selection coefficient for each category based on Torgerson et al. 2009 for noncoding mutations and Kim et al. 2017 for nonsynonymous mutations, mutation rate, target size (bp), expected number of mutations per genome (2*target size*mutation rate), and expected burden of new mutations (2*s*expected number of new mutations).
(XLSX)

**S5 Table. Fitness effects of inherited STR alleles.** For each individual tested, the table shows the number of variant STR alleles, number of STR alleles considered, total additive burden, and mean *s* per variant allele (see **Methods**). STRs from classes with poor model fits (KS score ≤ 50, see S2 Table) were excluded. We also present scaled burdens and mean allele effects that divide the original values by 2 to be comparable to SNV-based burden calculations (see **Methods**).
(XLSX)

**S6 Table. Fitness effects of inherited SNVs.** For each genomic annotation (rows), the table shows the number of variants and derived alleles (number of heterozygous genotypes + 2 * number of homozygous genotypes) carried by NA06985, NA06986, NA06994, HG02234, HG02237, and HG02240 (averaging over the 6 individuals). Mean fitness burden is computed as described in the **Methods**, by weighing each genotype by *s* for the relevant genotype and functional category. The mean *s* values represent the mean of the distribution of the segregating variants in the SLiM simulation.
(XLSX)

**S1 Fig. Evaluating whether the optimal allele matches the modal allele under different mutation and selection scenarios.** For each setting tested, we performed forward simulations of 2,000 STR loci using the procedure described in **Methods**. The x-axis gives the simulated selection coefficient and the y-axis gives the percent of simulations for which the modal allele matched the simulated optimal allele. Different lines each panel represent simulations performed under inferred best fit mutation parameters for different classes of STRs (see Fig 2). Left = dinucleotides, middle = trinucleotides, right = tetranucleotides. Colors denote different mutation models for each optimal allele length (e.g., "2_11" uses the model for dinucleotides with optimal allele length 11 repeats).
(TIF)

**S2 Fig. Distribution of the observed modal allele under different mutation and selection settings.** For each setting tested, we performed forward simulations of 2,000 STR loci using the procedure described in **Methods**. In each panel, the x-axis gives the inferred optimum (modal allele) and the y-axis gives the number of simulations. Gray histograms show the distribution of the inferred optimum (modal allele). Black vertical lines show the simulated optimum allele. Top row: $s = 0$, middle row: $s = 0.0001$; bottom row: $s = 0.1$. Left, middle, and right panels show simulations for dinucleotides with true optimum alleles of 11, 15, and 20, respectively.
(TIF)

**S3 Fig. Forward simulations demonstrate the effect of different parameters on allele frequencies.** (a) Effect of *L* (length-dependent mutation rate parameter): When $L = 0$, allele frequencies typically follow a bell-shaped curve, whereas when $L > 0$, allele frequencies tend to follow a bimodal distribution. (b-c) Effect of μ (per-generation mutation rate): Given

a value of *s*, as the mutation rate increases, the variability in the allele frequencies increases. When the mutation rate is low, it is difficult to distinguish between the effects of selection and low mutation at a single STR because there is little variability in the allele frequencies regardless of the selection coefficient. (d) Effect of β (mutation size directional bias): As β increases, so does the probability that an allele mutates toward the central allele. Thus, the resulting allele frequencies are more centered around the central allele and have lower variance. In each subplot, the x-axis gives allele size, in terms of the number of repeats away from the optimal allele (0). The y-axis gives the frequency of each allele in either the initial allele frequencies (left of the arrow) or forward simulated allele frequencies (right of the arrow). Results are based on representative replicates from running the forward simulation for $g = 20,000$ generations.
(TIF)

**S4 Fig. Detailed schematic of SISTR2.** First, we start with a list of observed allele frequencies across a set of STRs and set the mutation model parameters based on the STR class the loci belong to (e.g., dinucleotide STRs with optimal allele length 11 repeats). Next, we perform approximate Bayesian computation (ABC) by drawing 50,000 (*a*,*b*) gamma distribution parameters from the prior, drawing x *s* values from each gamma distribution, simulating allele frequencies forward in time for each *s*, and comparing the resulting heterozygosity distribution with the empirical distribution using a mean of differences distance metric. The top 1% of (*a*,*b*) parameters with the most similar heterozygosity distributions to the observed values are accepted. Values of (*a*,*b*) from accepted simulations are used to construct the posterior distribution on the mean, which is calculated as $s = a*b$. The (*a*,*b*) pair with the median mean is used to obtain an estimated distribution of *s*.
(TIF)

**S5 Fig. Comparison of SISTR (per-locus selection inference) vs. SISTR2 (joint estimation across loci) on simulated data.** In each plot, the x-axis denotes the selection coefficient, and the y-axis shows the fraction of STRs in each category with selection coefficients greater than the x-axis value. (a) Loci with a low baseline mutation rate. Unlike the per-locus method (red lines), the joint method (blue lines) can estimate *s* for short loci with a low baseline mutation rate, such as trinucleotides with an optimal allele of 5 and a baseline mutation rate of $10^{-5}$ [5]. (b) Loci with a high baseline mutation rate. The per-locus method is better at inferring *s* for loci with higher mutation rates (e.g., dinucleotides with an optimal allele of 11 and baseline mutation rate of $10^{-4}$.[25] as shown here) compared to loci with lower mutation rates. However, even at more highly mutable loci, the joint method is still more accurate than the per-locus method since it can distinguish between lower values of *s*. Overall, both the selection coefficient distributions obtained from the per-locus and joint methods are more concordant with the ground truth distribution as the mean *s* value increases.
(TIF)

**S6 Fig. Evaluation of SISTR2 selection inferences based on the number of input STR loci.** We tested SISTR2 under various conditions (left = short trinucleotides with low mutation rates, middle = short dinucleotides with moderate mutation rates, right = long trinucleotides with high mutation rates). For each condition, we evaluated SISTR2's selection inferences as a function of the number of STR loci used as input. In the top panels, the x-axis gives the simulated mean value of *s* and the y-axis gives the inferred mean value across 20 estimates. Colors denote the number of loci used for inference (red = 1, green = 10, orange = 100, blue = 1,000). Bottom plots show the mean +/- 1 s.d (purple) across the 20 simulations in each condition. True mean values of s are given by black horizontal lines. For all simulations shown, gamma distribution parameter *a* (i.e., the shape parameter) was set to 0.6.
(TIF)

**S7 Fig. Evaluating SISTR2 in the presence of genotyping errors.** (a) and (b) are the same as Fig 1b-c, except "observed" allele frequencies contained simulated STR genotyping errors (**Methods**). Our results demonstrate that except in cases with extremely low underlying mutation rates (e.g., short trinucleotides), modest genotyping errors do not bias

selection inferences. Short dinuc = optimal allele 11, long dinuc = optimal allele 20, short trinuc = optimal allele 5, long tri-nuc = optimal allele 13, short tetranuc = optimal allele 7, long tetranuc = optimal allele 10.
(TIF)

**S8 Fig. Robustness of SISTR2 to model misspecification and prior choice.** We evaluated SISTR2's performance in multiple settings. In each case, we simulated sets of 100 loci under 6 different true mean values of $s$ and determined the median inferred $s$ out of 10 simulations. In all cases true $s$ values were drawn from a gamma distribution with $a = 0.6$ and $b = \text{mean}(s)/a$. Baseline simulations are based on a dinucleotide model with an optimum allele of 11 (mutation model "e" from Fig 2). Black stars indicate the ground truth (simulated) $s$ value. Other lines show different scenarios to test the robustness of SISTR2 to differences in various inputs. Black = correct optimum and mutation model; Red = noisy optimum. In this case, 80% of input loci were simulated with the correct optimum allele (11 repeats) and for 20% of input loci the true optimum differed from 11 by a step size drawn from a geometric distribution with parameter $\rho = 0.5$. Steps had an equal probability to be insertions or deletions; Blue = noisy optimum. Same as red, but 20% of input loci had the correct optimum and 80% had incorrect optimums; Green = incorrect mutation model. Loci were simulated using model "e" for dinucleotides with optimum allele 11 but SISTR2 inference was performed assuming mutation model "a" for tetranucleotides with opti-mum allele 9; Yellow = extremely incorrect optimum. Loci were simulated with an optimum allele of 20 but SISTR2 infer-ence was performed assuming an optimum of 11; Purple = biased optimum. We simulated 20% of loci to have an optimum of 13 and 80% to have optima of 13 +/- step sizes drawn from a geometric distribution as in the red/blue lines. We then performed SISTR2 inference assuming an optimum of 11. This is meant to simulate cases where loci with longer optimum tend to have shorter modal alleles (see S2 Fig, right panels); Brown = alternative prior. Whereas the default prior draws the mean $s$ from a lognormal distribution with the mean and standard deviation of the underlying normal distribution set to log(0.0003) and log(30), respectively, in this case we set those to log(0.001) and log(10).
(TIF)

**S9 Fig. Assessing the goodness of fit of best-fit mutation models.** For each STR class, corresponding to a particular repeat unit (x-axis) and optimal allele (y-axis), we determined the mutation model with the best fit (Fig 2; Methods) and assessed the goodness of fit to observed data. For each class, we simulated 100 datasets based on the inferred mutation model, and compared the distribution of heterozygosities for simulated loci to those from observed loci using a KS test. Each heatmap cell shows the percent of simulation rounds with KS test p-value>0.05, indicating the two distributions are similar and the model fits well. Left = dinucleotides, middle = trinucleotides, right = tetranucleotides.
(TIF)

**S10 Fig. Assessing mutation model goodness of fit on subsampled data.** To reduce differences in power across different STR classes to detect differences between observed vs. simulated heterozygosity distributions, we repeated the goodness of fit analysis (S9 Fig) but including at most 50 STRs sampled from each class. Panels are the same as in S9 Fig. Left = dinucleotides, middle = trinucleotides, right = tetranucleotides.
(TIF)

**S11 Fig. Visualization of observed vs. simulated heterozygosity distributions.** For classes of loci with a low per-centage of simulations that fit the best-fit mutation model well, we plotted the observed vs. simulated heterozygosity distributions for a range of models to visually assess their similarity. The observed heterozygosity distribution is in black, the best-fit mutation model is in green, and other mutation models for comparison are in red/orange. Visually, the best-fit mutation model results in simulated heterozygosity distributions close to observed distributions, even for cases where the KS test indicated a poor fit.
(TIF)

**S12 Fig. Comparison of repeat unit-specific dinucleotide mutation rates inferred from SISTR2 versus _de novo_ mutations.** We compared the dinucleotide mutation rates inferred by SISTR2 (dashed lines) to those inferred from _de novo_ mutations based on WGS of the Simons Simplex Collection (SCC) dataset [8] (solid lines with points) and from _de novo_ mutations based on capillary electrophoresis from Icelandic individuals [16] (solid horizontal lines). The SISTR2 inferred mutation rate for each motif/optimal allele combination was set as the median mutation rate in the posterior of accepted mutation models from the feasible mutation parameters analysis (Fig 2). In all datasets, AT repeats have higher mutation rates than AC or AG repeats. Additionally, inferences from SISTR2 and in SSC show AC repeats mutate faster than AG repeats for shorter but not longer repeat tracts.
(TIF)

**S13 Fig. Goodness of fit assessments of the SISTR2 motif analysis scores.** For each class of loci (i.e., optimal allele/ repeat unit pair), 100 heterozygosity distributions were simulated using the distribution of selection coefficients inferred by SISTR2. Then, a KS test was used to compare the simulated distributions to the observed distributions. Each heatmap cell shows the percent of simulation rounds with KS test p-value>0.05, indicating the two distributions are similar and the model fits well. These values are reported as the KS Score in S2 Table. Left = dinucleotides, middle = trinucleotides, right = tetranucleotides.
(TIF)

**S14 Fig. Mean selection coefficient for each repeat unit and optimal allele length.** In each panel, the x-axis gives the optimal allele length and the y-axis gives the median inferred selection coefficient. Each color denotes a different repeat unit sequence. Categories computed based on fewer than 10 loci were excluded from this analysis. Panels (a), (b), and (c) show dinucleotides, trinucleotides, and tetranucleotides, respectively.
(TIF)

**S15 Fig. Mutation/selection parameters and expected burden of mutations for each STR class for _de novo_ mutations.** The x-axis shows different STR classes based on the repeat unit (annotated at the top) and optimal allele length. Optimal allele lengths range from 11-20 repeats (dinucleotides), 5–13 repeats (trinucleotides), and 7–10 repeats (tetranucleotides). Top panel: mean selection coefficient per mutation in each class. Second from top: number of STR loci considered in each class. Second from bottom: mean mutation rate for each class. Bottom: expected burden of _de novo_ mutations for each class. STR classes excluded from burden calculations due to low fit are colored in gray.
(TIF)

**S16 Fig. Burden of noninherited vs. standing variation in three children.** For each sample, we determined the selection coefficient for each nonoptimal inherited allele (red lines) (detailed procedure described in **Methods**) and each noninherited allele (which could arise due to _de novo_ mutation or cell line artifacts) (blue lines) using SISTR2. Noninherited alleles matching the optimal allele were excluded. Lines show the cumulative frequencies of selection coefficients for each category.
(TIF)

## Author contributions

**Conceptualization:** Kirk E. Lohmueller, Melissa Gymrek.

**Formal analysis:** Arun Durvasula, Nima Mousavi, Helyaneh Ziaei-Jam, Mikhail Maksimov.

**Methodology:** Bonnie Huang.

**Software:** Bonnie Huang.

**Supervision:** Kirk E. Lohmueller.

**Validation:** Bonnie Huang.

**Visualization:** Bonnie Huang, Melissa Gymrek.

**Writing – original draft:** Bonnie Huang, Kirk E. Lohmueller, Melissa Gymrek.

**Writing – review & editing:** Bonnie Huang, Kirk E. Lohmueller, Melissa Gymrek.

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
