## [Decision Letter · Decision Letter 0]

2 Jul 2025

PGENETICS-D-25-00338

Genome-wide selection inference at short tandem repeats

PLOS Genetics

Dear Dr. Gymrek,

Thank you for submitting your manuscript to PLOS Genetics. After careful consideration, we feel that it has merit but does not fully meet PLOS Genetics's publication criteria as it currently stands. Therefore, we invite you to submit a revised version of the manuscript that addresses the points raised during the review process.

Please submit your revised manuscript within 60 days Aug 31 2025 11:59PM. If you will need more time than this to complete your revisions, please reply to this message or contact the journal office at plosgenetics@plos.org. Please include the following items when submitting your revised manuscript:

We look forward to receiving your revised manuscript.

Kind regards,

Zachary A Szpiech

Guest Editor

PLOS Genetics

Kelly Dyer

Section Editor

PLOS Genetics

Aimée Dudley

Editor-in-Chief

PLOS Genetics

Anne Goriely

Editor-in-Chief

PLOS Genetics

**Additional Editor Comments :**

Thank you for your submission to PLOS Genetics. Two reviewers have read your paper and provided feedback. Both reviewers express enthusiasm for the work, but each has indicated several items that should be addressed. Please pay particular attention to comments 2 and 3 from reviewer 1. I agree that additional details on the ABC approach would improve reproducibility and that some consideration of model misspecification is warranted.

**Journal Requirements:**

At this stage, the following Authors/Authors require contributions: Bonnie Huang, Arun Durvasula, Nima Mousavi, Helyaneh Ziaei-Jam, Mikhail Maksimov, Kirk Lohmueller, and Melissa Gymrek. Please ensure that the full contributions of each author are acknowledged in the "Add/Edit/Remove Authors" section of our submission form.

The list of CRediT author contributions may be found here: https://journals.plos.org/plosgenetics/s/authorship#loc-author-contributions

https://journals.plos.org/plosgenetics/s/submission-guidelines#loc-parts-of-a-submission

5) We have noticed that you have uploaded Supporting Information files, but you have not included a list of legends. Please add a full list of legends for your Supporting Information files after the references list.

1) State what role the funders took in the study. If the funders had no role in your study, please state: "The funders had no role in study design, data collection and analysis, decision to publish, or preparation of the manuscript.".

**Reviewers' comments:**

Reviewer's Responses to Questions

Reviewer #1: Huang et al present SISTR2, a computational method that is designed to infer the distribution of selection coefficients for a set of short tandem repeats (STRs) based on their allele frequencies. STRs are implicated in some heritable diseases and have long been of interest in population genetics studies, but most methods for inferring selection focus on single nucleotide variants. I found this to be a nicely written and (overall) clear manuscript that makes a novel contribution to the population genetics literature. I do have a few comments and questions for the authors.

Main comments:

1. The overall logic of the method makes sense -- given an optimal allele and a set of STRs, selection will constrain the distribution of heterozygosities relative to a mutation-only model. As the authors note, inferring a mutation rate model is hence a critical step in the inference. Is it not also true that a model in which mutation rates vary across loci could explain some of the variation in heterozygosities across loci? It was not clear to me if/how mutation rate variation across loci (within a single class of STRs defined by the repeat unit) was considered in the study. As I understood it, a single best-fitting mutation rate model was fit for each class of loci (based on sequence content and repeat length), and the selection inference was performed on the basis of this mutation model. Is there a biological reason to think that mutation rate might vary across loci, and if so how would this affect the selection inference?

2. Description of ABC method: The approach uses ABC (approximate Bayesian computation) to infer the distribution of selection coefficients. I think this is a great approach and a nice contribution. While I was able to follow the logic of the ABC method in broad strokes, several questions remained for me after reading through the methods. I will highlight my main questions here, but my general request is that the authors revise this section to improve the reproducibility.

2a. The method uses observed heterozygosity across a set of STRs as the summary statistic of interest in ABC. There are other statistics computed from the frequency distributions that could have been considered. Why was this statistic chosen? Were others considered, and is there any reason to think that this should be among the most informative possible choices?

2b. The text states that the method "Compare(s) the distribution of the simulated heterozygosity for all loci within the set to the distribution of the observed heterozygosity for all loci in the set by sorting both vectors, obtaining the difference vector, and taking the mean of the absolute value of the differences." It sounds like this difference vector consists of the per-locus heterozygosities for each locus? If so, in my opinion this choice needs a little more justification. It seems like a few outlier loci in either the observed or simulated data could have a disproportionately large impact on the deviation between model and observed. Most ABC approaches summarize the observed/modeled data with a few informative statistics to reduce the dimensionality of the difference calculation.

2c. The method assumes that the modal allele is the fittest allele, and in the Discussion it is said that "<our method=" "> must analyze STRs with different optimal alleles separately". If I understand correctly, the authors pool together STRs with the same repeat unit (e.g., "AAAT" repeats), and in each round of selection inference include only alleles with a given observed modal allele. Is this correct? Or do they combine all repeats with a given repeat unit, and assume that a single fittest allele based on the mode across all loci in the set?

3. Validation of ABC

3a. If it is true that only loci with the same modal allele are used in the inference step, how does this assumption play into validation? When selection is strong, I would expect that the modal allele is typically the fittest. When selection is weak, I would suppose that the modal allele is not always the fittest allele. Have the authors considered how this might affect inference?

3b. In general, the manuscript would benefit from some consideration of model misspecification. I don't think it is necessary to perform a sweep across many sets of parameters, but checking how assumptions about mutation models/symmetry, demography, etc in a couple of basic scenarios would be useful for assessing the robustness of the method.

4. Empirical literature on STR mutation rates

The manuscript does a nice job of summarizing other population genetics studies of mutation rates and selection on STRs and putting this study into context, but there is also quite a lot of molecular genetics research on STRs. I'm not an expert on this literature, so I'm not going to suggest specific papers, but in general the paper would benefit from more linking back to the empirical literature on STRs. What is known (or at least previously studied) for STR mutation rates from pedigrees or other studies? How do these mutation rate estimates compare to those in this study?

Minor comments:

It's not entirely clear to me why we can assume that the empirical STR loci (genotyped with GangSTR and used to infer mutation models) evolve neutrally — this could be clarified.

I was confused by the statement "The model with the highest number of ABC acceptances corresponds to the maximum likelihood mutation model" (and other references to maximum likelihood). Maximum likelihood is a different statistical approach from ABC which depends on computing a likelihood function, and in general model comparison is one of the things that is not straightforward to do with ABC. I think it's fine to allow the model with the most acceptances to be the "best-fitting" for the purposes here, but I don't think it should be called a maximum likelihood fit unless I have misunderstood the approach.

The Discussion notes that the mutation model is symmetric — this was not clear to me earlier in the manuscript and (unless I missed it) this information should be stated earlier.

The discussion states "Further, it is not immediately clear why different STR loci with the same repeat unit would have different optimal lengths." I don't think it's too hard to think of models where this isn't true, such as a model where the selective effect is only due to STR length and the optimal length is dependent on local sequence context and not the specific base composition of the STR? If there is other literature supporting this assertion a citation would be helpful, otherwise I think the opposite point could also be claimed just as easily.</our>

Reviewer #2: The authors present a new ABC method to jointly infer the distribution of fitness effects and mutational model of a set of STR mutations. The authors first show that their method correctly estimates parameters from the distribution of fitness effects via a set of simulations. Then, the authors show how well the method can infer the mutational model of a set of presumed neutral STR mutations. Then, the authors apply their method to estimate the distribution of fitness effects of STRs with different classifications that depend on their functional annotation due to being on a particular region (“coding”, “intron”, “promoter”, etc. ) and also due to their particular motif. Finally, the authors interestingly show that the mutational burden of STR variants is bigger than the mutational burden of SNPs.

I think that the authors did a great job explaining their method and that the conclusions are very relevant to understand the functional importance of STR variation. I have a few major comments on the manuscript that are mostly focused on the assumptions behind the method and a few minor comments.

Major comments

“Our mutation model is based on a generalized stepwise mutation (GSM) model with two modifications, including a length-dependent mutation rate and a directional bias in mutation sizes toward an optimal (central) allele length”

Is there a reason behind the selection of that model? If there is, it would be good to provide more information in the text. Can the authors provide some discussion on how the prior distribution of the mutational model chosen has an impact on their conclusions?

Can the authors provide some guidelines or discussion regarding how to choose an appropriate prior distribution for all the parameters in the analyses presented?

“Indeed, recent work has suggested that the modal allele is the most influential summary statistic at inferring the optimal allele and in cases where it was possible to infer to infer the optimal allele, the optimal allele was typically the modal allele (in single optima models) or a multiple of the optimal allele (in periodic optima models)6.”

I think this is a reasonable assumption but surely this depends on the fitness difference between the optimal STR allele and the other alleles. If the difference in fitness between the optimal allele and all the other alleles, in terms of Ns values, is larger than 1 then likely the modal allele will be the optimal allele. However, if the difference, in terms of Ns values, is smaller than 1 then likely the modal allele won’t always be the optimal allele.

Could the authors provide more information that backs up the presented assumption? One possibility could be to do simulations under different Ns values and show that the modal allele is the optimal allele under different differences of Ns values between the optimal allele and all the other alleles. I am also open to other possibilities such as discussing more information from references that back up this assumption.

Minor comments

Figure 1a) The caption does not describe what the blue variables mean.

Figure 2.- I would recommend adding more information in the caption about what the models a-f mean.

I am also unsure about why the Log10 mutation rate varies as a function of the repeat length. Is the repeat length an average of the repeat length across many STRs? Can the authors provide more information behind this behavior?

“For tetranucleotides, AAGG, AAAG, and AGAT, which also showed higher mutation rates than other tetranucleotide STRs13 (Fig. 2f),”

This is hard to see quickly in the figure. Can more information be provided that backs up this statement?

It would be good to show the compared “Total burden” on Table S5 and Table S6 to compare the total burden between SNPs and STR variants.

“ is mutation rate of each allele x is determined as is the length-dependent mutation rate, such that the (µ ) = (µ0 ) + ; β is the length constraint, indicating the bias of long alleles to contract and short alleles to expand toward the optimal allele length; ρ is the step size parameter describing the geometric distribution from which mutation step sizes are drawn.”

Can you explain in mathematical terms how β and ρ work?

I might have missed that in the text but how do you define the “presumed neutral loci”?

**Have all data underlying the figures and results presented in the manuscript been provided?**

Reviewer #1: **No: ** Links to the summary statistics used in the study would be more helpful than only listing the SRA accession.

Reviewer #2: Yes

PLOS authors have the option to publish the peer review history of their article (what does this mean? ). If published, this will include your full peer review and any attached files.

**Do you want your identity to be public for this peer review?** For information about this choice, including consent withdrawal, please see our Privacy Policy .

Reviewer #1: **Yes: ** Lawrence Uricchio

Reviewer #2: No

**Figure resubmission:**
---

## [Decision Letter · Decision Letter 1]

17 Nov 2025

Dear Dr Gymrek,

We are pleased to inform you that your manuscript entitled "Genome-wide selection inference at short tandem repeats" has been editorially accepted for publication in PLOS Genetics. Congratulations!

Yours sincerely,

Kelly A. Dyer

Section Editor

PLOS Genetics

Aimée Dudley

Editor-in-Chief

PLOS Genetics

Anne Goriely

Editor-in-Chief

PLOS Genetics

BlueSky: @plos.bsky.social

Comments from the reviewers (if applicable):

Reviewer's Responses to Questions

**Comments to the Authors:**

Reviewer #1: I thank the authors for their thorough consideration of my previous questions and their work to further validate/justify the ABC approach. I think this is a great paper. I have no further comments or questions at this time.

Sincerely,

Lawrence Uricchio

Reviewer #2: The authors have written a comprehensive response to my comments. I consider that the paper has improved substantially thanks to the careful work by the authors. This paper is an important contribution to the field.

**Have all data underlying the figures and results presented in the manuscript been provided?**

Reviewer #1: Yes

Reviewer #2: None

PLOS authors have the option to publish the peer review history of their article (what does this mean? ). If published, this will include your full peer review and any attached files.

**Do you want your identity to be public for this peer review?** For information about this choice, including consent withdrawal, please see our Privacy Policy .

Reviewer #1: **Yes: ** Lawrence Uricchio

Reviewer #2: No

**Data Deposition**

http://datadryad.org/submit?journalID=pgenetics&manu=PGENETICS-D-25-00338R1

**Press Queries**

---

## [Editor Report · Acceptance letter]

PGENETICS-D-25-00338R1

Genome-wide selection inference at short tandem repeats

Dear Dr Gymrek,

We are pleased to inform you that your manuscript entitled "Genome-wide selection inference at short tandem repeats" has been formally accepted for publication in PLOS Genetics! Your manuscript is now with our production department and you will be notified of the publication date in due course.

With kind regards,

Anita Estes

PLOS Genetics

On behalf of:
